# Lysosomal Changes in Mitosis

**DOI:** 10.3390/cells11050875

**Published:** 2022-03-03

**Authors:** Jonathan Stahl-Meyer, Lya Katrine Kauffeldt Holland, Bin Liu, Kenji Maeda, Marja Jäättelä

**Affiliations:** 1Cell Death and Metabolism, Center for Autophagy, Recycling and Disease, Danish Cancer Society Research Center, 2100 Copenhagen, Denmark; lyho@cancer.dk (L.K.K.H.); liu@cancer.dk (B.L.); kenjim@cancer.dk (K.M.); 2Department of Cellular and Molecular Medicine, Faculty of Health Sciences, University of Copenhagen, 2200 Copenhagen, Denmark

**Keywords:** lysosome, cell cycle, mitosis, lysosomal leakage, lipidome

## Abstract

The recent discovery demonstrating that the leakage of cathepsin B from mitotic lysosomes assists mitotic chromosome segregation indicates that lysosomal membrane integrity can be spatiotemporally regulated. Unlike many other organelles, structural and functional alterations of lysosomes during mitosis remain, however, largely uncharted. Here, we demonstrate substantial differences in lysosomal proteome, lipidome, size, and pH between lysosomes that were isolated from human U2OS osteosarcoma cells either in mitosis or in interphase. The combination of pharmacological synchronization and mitotic shake-off yielded ~68% of cells in mitosis allowing us to investigate mitosis-specific lysosomal changes by comparing cell populations that were highly enriched in mitotic cells to those mainly in the G1 or G2 phases of the cell cycle. Mitotic cells had significantly reduced levels of lysosomal-associated membrane protein (LAMP) 1 and the active forms of lysosomal cathepsin B protease. Similar trends were observed in levels of acid sphingomyelinase and most other lysosomal proteins that were studied. The altered protein content was accompanied by increases in the size and pH of LAMP2-positive vesicles. Moreover, mass spectrometry-based shotgun lipidomics of purified lysosomes revealed elevated levels of sphingolipids, especially sphingomyelin and hexocylceramide, and lysoglyserophospholipids in mitotic lysosomes. Interestingly, LAMPs and acid sphingomyelinase have been reported to stabilize lysosomal membranes, whereas sphingomyelin and lysoglyserophospholipids have an opposite effect. Thus, the observed lysosomal changes during the cell cycle may partially explain the reduced lysosomal membrane integrity in mitotic cells.

## 1. Introduction

Lysosomes, late endosomes and other closely related acidic organelles (hereafter together referred to as lysosomes) are the major recycling stations of the cell. In their acidic lumen, they employ more than 60 acid hydrolases to degrade and recycle intra- and extracellular macromolecules that are received via endocytosis and autophagy [1,2,3]. Besides providing the cells with energy sources and building blocks for new macromolecules, lysosomes act as central regulators of cellular metabolism and growth by sensing cellular nutrient availability and adapting the metabolism accordingly [2,4]. Such regulatory signaling is mediated by a network of proteins residing on the lysosomal outer membrane. For example, mammalian target of rapamycin (mTOR/*MTOR*) complex 1 (mTORC1) assisted by vacuolar-type ATPase (V-ATPase), lysosomal amino acid transporter SLC38A9, and numerous mTORC1-associated proteins, activates anabolic signaling cascades in response to high arginine or cholesterol concentrations in the lysosomal lumen [4,5,6,7,8], while AMP-activated protein kinase (AMPK) activates autophagy and other catabolic pathways when nutrients are scarce [9,10].

In addition to serving as a metabolic signaling hub, the lysosomal limiting membrane constitutes an important boundary between the cytosol and the degradative enzymes in the lysosomal lumen. Thus, maintaining the integrity of this membrane has been long considered as an absolute prerequisite for cell viability [11,12,13,14]. The abundant and heavily glycosylated integral membrane proteins, such as lysosomal-associated membrane proteins 1 and 2 (LAMP1/*LAMP1* and LAMP2/*LAMP2*), protect the lysosomal limiting membrane against lysosomal membrane permeabilization that is induced by lysosomal lipases and drugs targeting the lysosomal membrane [15,16]. Instead of the glycosylated limiting membrane, luminal lipases are recruited to intraluminal vesicles (ILVs) through interactions with bis(monoacylglycero)phosphate (BMP) and other negatively charged phospholipids that are found on the surface of these vesicles [17,18]. At this site, lysosomal lipases degrade all sorts of intra- and extracellularly derived lipids, whose degradation products are exported to the cytosol by luminal, transmembrane, and cytosolic lipid transporters, such as Niemann-Pick type C1 (NPC1/*NPC1*), Niemann-Pick disease type C2 (NPC2/*NPC2*), and Oxysterol-binding protein-related proteins [19]. An imbalance in the tightly ordered processes of lipid metabolism within the lysosomes has serious consequences, as exemplified by numerous genetic diseases that are caused by mutations in the genes controlling lysosomal lipid metabolism or transport as well as the ability of cationic amphiphilic drugs (CADs) and other pharmacological inhibitors of lysosomal lipases to trigger lysosomal membrane permeabilization and lysosome-dependent cell death in cancer cells [20,21]. CAD-induced lysosomal membrane permeabilization and subsequent leakage of luminal cathepsins to the cytosol depend on the effective inhibition of acid sphingomyelinase (ASM/*SMPD1*) and correlate with the accumulation of sphingomyelin and lysoglyserophospholipids in lysosomes [22,23].

Besides the drug-induced lethal lysosomal leakage, which is emerging as an attractive anti-cancer strategy, a spatiotemporal form of lysosomal leakage has recently been suggested to regulate normal cellular processes, such as inflammation, cell adhesion, motility, and mitosis [24]. In the majority of prometaphase cells, a few chromatin-proximal lysosomes momentarily lose their membrane integrity and the proteolytic activity of released cathepsin B (CTSB/*CTSB*) promotes accurate chromosome segregation in the anaphase [25]. The mechanisms controlling mitotic lysosomal leakage remain elusive, but changes in lipid homeostasis may be among the underlying factors. This is supported by data showing that cholesterol supplementation counteracts the lysosomal leakage in the prometaphase and metaphase and increases the frequency of chromosome segregation errors in anaphase [25]. Also, other perturbations in cellular lipid metabolism and trafficking have been reported to cause problems in cytokinesis and an accumulation of binucleated cells [26,27].

When cells enter mitosis, most cellular processes that are not directly participating in the cell division, e.g., transcription [28,29], translation [30,31], and autophagy [8,32], are shut down or strongly downregulated. Endolysosomal compartment remains, however, highly active during mitosis and has at least two other mitosis-specific tasks besides the above-described control of chromosome segregation [25,33,34]. Endocytosis decreases the plasma membrane surface area and rounds the cell shape in the prophase, whereas lysosomal exocytosis donates membrane to restore the plasma membrane surface area once the cell division is completed [35,36,37,38]. In line with the dramatic changes in membranes, cellular lipid homeostasis is significantly altered during mitosis [27]. How the composition and activity of lysosomes change during mitosis remains, however, largely uncharted. Such knowledge is crucial to understand the molecular background of mitotic lysosomal leakage. In order to study mitotic lysosomes, we optimized here methods to enrich U2OS osteosarcoma cell cultures for mitotic cells and to purify lysosomes from these enriched cultures. For this purpose, we used a reversible cyclin-dependent kinase 1 (CDK1/*CDK1*) inhibitor to arrest the cells at the G2 phase of the cell cycle and then released the cells to enter mitosis. Cell cultures that were enriched for the G2 phase and mitosis and untreated cells presenting all cell cycle phases were then analyzed for various lysosomal and lysosome-associated proteins by immunoblotting and for lysosomal size and pH by microscopy. Finally, lysosomes that were purified from these three culture conditions were analyzed by mass spectrometry (MS)-based shotgun lipidomics to quantitatively determine their lipid composition. These analyses revealed significant mitosis-specific lysosomal changes in size, pH, and composition that may contribute to the decreased stability of mitotic lysosomes.

## 2. Material and Methods

### 2.1. Cell Culture and Treatments

U2OS human osteosarcoma cells (ATCC^®^ HTB-96^TM^) were obtained and authenticated by the American Type Culture Collection (ATCC, Danvers, MA, USA). The cells were cultured in DMEM high glucose Dulbecco’s modified Eagle’s medium (DMEM, Thermo Fisher Scientific, Waltham, MA, USA, #31966-021) that was supplemented with 10% fetal calf serum (FCS, Life Technologies, Carlsbad, CA, USA, #10270-106) and 1X penicillin/streptomycin (Life Technologies, #15140-122).

### 2.2. Cell Synchroniztion and Release

To synchronize the cells in the late G2 phase, the cells were treated with 9 µM RO-3306 (Merck Millipore, Darmstadt, Germany, #217721) for 16 h at 37 °C in a humidified atmosphere of 5% CO_2_. To release the cells into mitosis, the cells were washed twice for 10 min with pre-warmed (37 °C) culture medium and allowed to enter the required phase of mitosis for 30–60 min.

### 2.3. Collection of Cells

The isolation of untreated and synchronized cells was performed by first removing the detached dead cells by washing with cold PBS before scraping and washing them once more. The isolation of mitotic cells was performed by washing the cells once in cold PBS before mitotic shake-off by knocking the dish into a hard surface. The mitotic cells were collected and spun down at 500× *g* for 5 min and washed twice in cold PBS.

### 2.4. Immunocytochemistry

The cells were grown on glass coverslips and fixed in 4% paraformaldehyde (PFA, Ampliqon, Odense, Denmark, #43226.1000). For LAMP2 stainings, the cells were incubated for 15 min with NH_4_Cl to quench the PFA signal before being permeabilized with ice-cold MeOH for 10 min. The cells were then permeabilized and blocked in Dulbecco’s phosphate-buffered saline (DPBS, Thermo Fisher Scientific, #14190-094) containing 5% goat serum (DAKO, USA, #X090), 1% (*w*/*v*) bovine serum albumin (BSA, Amresco, Radnor, PA, USA, #E531), and 0.3% (*v*/*v*) Triton X-100 (Sigma-Aldrich, St. Louis, MO, USA, #T9284). The cells were then incubated with the anti-p-H3, anti-LAMP2, and/or anti-tubulin primary antibody and matching 488- or 568-Alexa Fluor^®^-conjugated secondary antibody (Appendix A). The nuclei were stained with 1.25 μg/mL Hoechst-33342 (Sigma-Aldrich, #B2261). Coverslips were mounted with Prolong Gold Antifade mounting medium (Life Technologies, #P36930) and left to dry overnight prior to microscopy. The images were acquired on a LSM700 confocal microscope using a plan-Apochromat 63×/1.40 Oil DIC M27 objective and Zen 2010 software (6.0.0.485) (Carl Zeiss, Jena, Germany). Cell cycle profiling was carried out on a ScanR modular high-content screening station microscope using a UPLXAPO 20×/0.8 air objective. The acquired images were analyzed by FIJI (Java 1.8) or ScanR analysis software (3.2.0) and the illustrations were carried out in Spotfire software (11.4.1), Prism Software (8.0.2) or Adobe Illustrator (26.0.3).

### 2.5. Pierce BCA Assay

The protein concentrations were measured by a Pierce^TM^ BCA assay kit (Thermo Fisher Scientific, #23225) according to the manufactures protocol. The protein concentrations of the samples were calculated based on a BSA standard curve.

### 2.6. Western Blotting

The proteins in the cell lysates were separated in 4–15% Mini-PROTEAN TGX Gels (BIO-RAD, Hercules, CA, USA #456-1083) or midi-CRITERION TGX gels (BIO-RAD, #5671084), transferred to nitrocellulose membrane (BIO-RAD, #170-4158 and #170-4159) using Bio-Rad trans-blot turbo system, and stained with indicated primary antibodies and matching secondary HRP-conjugated antibodies (Appendix A). The immunoreactivity was detected with luminescent image reader (Fujifilm, Cambridge, MA, USA LAS-4000) following incubation with either clarity Western enhanced chemiluminescent (ECL) reagents (Bio-Rad, #170-5061) or ultra-sensitive ECL HRP substrate (Thermo Fisher Scientific, #PI34096), and the band intensity was quantified using FIJI software (Java 1.8).

### 2.7. Cathepsin Enzyme Assays

CTSD activity was measured using a CTSD activity kit (Abcam, Cambridge, UK, #ab65302) according to the manufactures protocol starting out with one million U2OS cells of each different population. For CTSB activity, one million U2OS cells of each different population were lysed in 500 µL lysis buffer (25 mM Hepes, 5 mM MgCl_2_, 1mM EGTA, 0.5 mM pefablock, 0.1 mM dithiothreitol (DTT), and 0.025% Triton X-100; pH 6.0)) and incubated for 15 min on shaking table at 4 °C. The lysates were then spun down at 3000× *g* at 4 °C for 15 min. A total of 50 µL lysates were transferred to a black bottom 96-well plate in triplicates together with 50 μL substrate buffer (cathepsin reaction buffer at pH = 6, 8 mM DTT, 0.5 mM pefablock, and 20 mM zFR-AFC (Sigma-Aldrich, P4157)). Lastly, light emission (excitation at 400 nm; emission at 505 nm) was measured on a SpectraMax ID3 multimode plate reader (Thermo Fisher Scientific) every 60 s for 30 min at 37 °C. The output values from both cathepsin assays were normalized by protein content.

### 2.8. pH Measurements

A total of 20,000 U2OS cells were seeded in an 8-well slide and cultured for 24 h. Next, the cells were treated for 16 h with 1.25 mg/mL fluorescein- and tetramethylrhodamine-coupled dextran (Thermo Fisher scientific, #D1951) to allow loading of the lysosomes by endocytosis. The cells were then washed and fresh medium was added for a three-hour chase period before being left untreated or treated with 100 nM concanamycin A (Santa Cruz, Dallas, TX, USA, #sc-202111) for 2 h. The medium was aspirated and the imaging solution was added (Thermo Fisher scientific, #A14291DJ) together with 625 ng/mL cell permeable Hoechst-33342 (Sigma-Aldrich, #B2261). The images were acquired on a LSM800 confocal microscope using a plan-Apochromat 63×/1.40 Oil DIC M27 objective and Zen 2010 software (Carl Zeiss). Data was analyzed in Excel and illustrations made in Prism Software (8.0.2) and Adobe illustrator.

### 2.9. FeDEX-Mediated Lysosomal Isolation

The lysosomes were purified according to [39] with minor applied changes. In brief, the cells were treated with FeDEX for 16 h and washed once with pre-warmed medium (37 °C) before being incubated for approximately 1 h. The cells were then collected as described above. Next, the cells were washed once in SuMa^4^ buffer (SuMa buffer (10 mM HEPES, 70 mM sucrose, 210 mM mannitol) that was supplemented with 0.5 mM DTT, 25 mg/mL fatty acid-free BSA, a tablet of Mini Complete protease inhibitor without EDTA (Sigma-Aldrich, #5056489001), and ≤1.25 units benzonase (Sigma-Aldrich, #E1014-25KU)), resuspended and lysed in 1 mL SuMa^4^ buffer by syringing the cells through a 25-gauge needle. This is referred to as “whole cell sample/fraction”. The lysates were centrifuged at 1000× *g* for 10 min, and the supernatant was transferred to a new tube, and the step was repeated. Next, the lysates were loaded onto equilibrated magnetic columns (Miltenyi Biotec, Bergisch Gladbach, Germany, #130-042-401) that were mounted in a magnetic rack, and washed in SuMa^4^ once, and three times in SuMa buffer. The columns were then removed from the magnetic rack and the bound material was eluted in SuMa buffer. The resulting eluate is referred to as “lysosomal sample/fraction”. All the samples were then centrifuged at 21,100× *g* for 20 min at 4 °C, and resuspended in 155 mM NH₄HCO₃.

### 2.10. LAMP1 Immunoprecipitation

The cells were harvested as described above, washed twice in SuMa buffer, and spun down at 1000× *g* for 5 min at 4 °C. The cells were then lysed in 1 mL SuMa buffer^4^ by syringing the cells through a 25-gauge needle. The lysates were centrifuged at 1000× *g* for 10 min, the supernatant was transferred to a new tube, and the step was repeated. Following, the lysates were mixed with either 0.6 µg/mL IgG control (Invitrogen, Waltham, MA, USA, #31235) or anti-LAMP-1 primary antibody (Abcam, #ab24170) and incubated for 30 min in a rotator at 4 °C. 25 µL prewashed microbeads (Miltenyi Biotec, #130-048-602) were then added to the mixtures and they were incubated for 60 min at 4 °C in a rotator. The samples were then loaded onto equilibrated magnetic columns (Miltenyi Biotec, #130-042-401) that were mounted in a magnetic rack and washed in SuMa^4^ once and three times in SuMa^2^ buffer (SuMa^4^ buffer without benzonase and a protease inhibitor tablet). The columns were then removed from the magnetic rack and the bound material was eluted in SuMa^2^. All the samples were then centrifuged at 21,100× *g* for 20 min at 4 °C, washed once SuMa buffer that was supplemented with 0.5 mM DTT, and spun down again 21,100× *g* for 20 min at 4 °C.

### 2.11. Lipid Extraction and Lipidomics ANALYSIS

Mass spectrometry-based quantitative shotgun lipidomics was performed according to a protocol from Nielsen and colleagues [40] with minor applied changes. In brief, lipid internal standards were added to each sample prior to lipid extraction to allow lipids to be quantitated. For a complete list of the lipid internal standards that were used in the present study, see Appendix A. A single step lipid extraction was performed with a modified Bligh and Dyer protocol. The lipid extracts was mixed with either positive (13.3 mM ammonium bicarbonate in 2-propanol) or negative (0.2% (*v*/*v*) tri-ethyl-amine in chloroform:methanol 1:5 (*v*/*v*)) ionization solvents. The lipidomics analysis was performed on a quadrupole-Orbitrap mass spectrometer Q Exactive that was equipped with a TriVersa NanoMate to allow a direct and automated infusion of the sample. The data acquisition cycle consisted of MS and MS/MS scans in both positive and negative modes. The ion spectra was processed using a Python-based software, LipidXplorer [41]. LipidXplorer reported the identified lipid species as well as their *m/z* values and the intensities of associated precursor and fragment ions. A subsequent sorting and quantification of lipid species were achieved using an in-house built R-based suite of scripts named LipidQ (https:/github.com/ELELAB/lipidQ). The absolute molar quantities were calculated based on intensity values of sample-derived lipids and their respective internal lipid standards.

### 2.12. Lipidomics Data Analysis and Statistics

The lipid levels were expressed as molar quantity relative to the sum of the molar quantities of all the identified lipid species in the sample (mol% of total) unless otherwise specified. A single lipid species was only included in the further analyses if the determined median quantity of all the replicates within a sample type was above 0.0001 mol%. The lipid profiles that were presented in heatmaps were visualized using the “pheatmap” package [42] in R (version 4.0.3.) and with unsupervised hierarchical clustering with Euclidian distance. For the differential expression of the lipids, the “limma” package [43] was used to fit a linear model to each lipid species based on the determined lipid quantities in the samples. Based on the models, log2-transformed fold change values and associated *p*-values were calculated. Lipid species were considered to have statistically significantly different quantities if the *p*-value was below 0.05. Unpaired *t*-tests were used for comparing the means of two groups, e.g., released vs. untreated samples. Lipid nomenclatures were as previously used [42]. Statistics on data other than lipidomics were analyzed with GraphPad Prism Software (8.0.2). Statistical analysis between two groups were determined using the either paired or unpaired two-tailed *t*-test depending on the data in question. *p*-values are shown in respective figures.

### 2.13. Acridine Orange

U2OS cells that were cultured in 35 mm glass-bottomed dishes (14 mm, No.1.5, MatTek Corporation, Ashland, MA, USA) were treated with 2.5 µM acridine orange (Thermo Fisher Scientific, #A1301) for 15 min at 37 °C to label lysosomes. The medium was aspirated and fresh growth medium was added prior to time-lapse imaging, which was performed in a heated chamber (37 °C) using a Plan-Apochromat 63×/1.4NA (Carl Zeiss) with differential interference contrast oil objective that was mounted on an inverted Zeiss Axio Observer Z1 microscope (Marianas Imaging Workstation from Intelligent Imaging and Innovations Inc. (3i), Denver, CO, USA), that was equipped with a CSU-X1 spinning-disk confocal head (Yokogawa Corporation of America, USA) and three laser lines (488 nm, 561 nm and 640 nm). The images were acquired using an iXon Ultra 888 EM-CCD camera (Andor Technology, Belfast, UK). The images were acquired every second for 180 s. The lysosomal volume and integrity as well as membrane integrity was evaluated by assessing the green and red fluorescence.

## 3. Results

### 3.1. CDK1 Inhibition Followed by Mitotic Shake-Off Leads to Mitotic Enrichment

Most cell lines in culture have only approximately 2% of cells in mitosis at all times [44,45]. In order to investigate if and how lysosomes alter their structural features and functionality during mitosis, we first established a reliable protocol to prepare a cell population that was highly enriched for mitotic cells. We selected U2OS cells due to their high proliferative rate and reduction in adherence when rounding up during mitosis. The former allowed for efficient synchronization by chemically arresting the progression of cell cycle and the latter enabled the efficient separation of the detached mitotic cells from those in the interphase that were adhered to the surface of cell culture dishes. We cultured U2OS cells for 16 h in the presence of reversible CDK1 inhibitor, RO-3306, to arrest them at the late G2 phase [46,47]. After 16 h, we either kept the cells arrested by continuing to culture them with RO-3306 for 1 h or released them from the arrest to allow them to enter mitosis by culturing them in a drug-free medium for 1 h (Figure 1a). We then evaluated the vehicle-treated control (hereafter referred to as untreated), arrested (hereafter referred to as synchronized) cells, and the released cells under a microscope after staining them for DNA (Hoechst) and phospho-S10-histone H3 (p-H3) that is a specific marker of mitosis from prophase to late anaphase/early telophase [48] (Figure 1b). We plotted the acquired mean Hoechst intensity against the total p-H3 intensity and defined four regions that, when gated, allowed the categorization of the cells into four separate populations; interphase cells, p-H3-negative mitotic/apoptotic cells, p-H3-positive interphase/prophase cells, and mitotic cells (Figure 1b). For this study, mainly the interphase (Figure 1b, blue gate) and mitotic (Figure 1b, red gate) gates were of interest. The total Hoechst intensity, shown as binned histograms, displays the classical linear cell cycle profile, in which the DNA content of a cell is proportional to its current cell cycle phase (Figure 1c). In other words, a cell in the G2 phase has twice the DNA content of a cell in the G1 phase. G2 phase and mitotic cells cannot be discriminated by this approach alone because they have the same DNA content. To circumvent this, pH3-positive mitotic cells were included in the plot by introducing a binary two-color code indicating cells in- (red) or outside (blue) the mitotic gate (Figure 1c). The untreated populations were enriched in non-mitotic cells with low DNA content (low Hoechst intensity), in accordance with the cells in the culture mainly being in the G1 phase [49,50]. In comparison, the synchronized cells had a larger population of cells with a high DNA content (Figure 1c), confirming successful arrest at the G2 phase. The released cells had a similar distribution of DNA content as the synchronized cells, but 15.4% (±4.69%) of them were mitotic in comparison to 1.3% (±0.26%) and 2.6% (±1.16%) of the untreated and synchronized cells, respectively (Figure 1d). To achieve an even higher enrichment of mitotic cells, we extended the protocol with mitotic shake-off of the released cells [51]. Flow cytometry analysis confirmed that approximately 69% of the released cells that were harvested with mitotic shake-off were in mitosis, in comparison to around 2% and 9% of the untreated and synchronized cells that were detached from the surface by trypsinization, respectively (Figure 1e,f). Taken together, our protocol extended with mitotic shake-off improved the yield of mitotic cells over four-fold.

### 3.2. The Expression of LAMP1 and CTSB Is Down-Regulated in Mitotic Cells

Moving forward in our attempt to reveal the lysosomal features that are unique to mitotic cells, we compared the released population highly enriched in mitotic cells to the untreated and synchronized cells, which represent populations that were enriched for cells in two different interphase cell cycle stages (the G1 and G2 phases, respectively), as controls. As these two controls represented untreated and RO-3306-treated cells, their use also prevented misinterpretation of possible side effects of RO-3306 as cell cycle-related changes. We first investigated the levels of various lysosomal proteins in the above-described population using Western blotting. Western blot analyses confirmed the elevated expression of p-H3 in the released cells and thus the successful enrichment of mitotic cells (Figure 2a). We then examined the selected core lysosomal proteins and found limiting membrane embedded LAMP1 and the catalytically active forms of luminal CTSB significantly reduced in the released cells compared to the untreated cells. When compared to the synchronized cells, the released cells still had significantly less mature CTSB, while the reduction in LAMP1 levels did not quite reach significance (*p* = 0.076) (Figure 2a). Similar to LAMP1, the ASM protein levels in the released cells were significantly reduced as compared to the untreated cells, but only a downward trend was observed when the released cells were compared with the synchronized cells (Figure 2a). Furthermore, the released cells demonstrated strong tendencies towards reduced levels of other lysosomal proteins that were analyzed, including LAMP2, cathepsin D (CTSD/*CTSD*), and B subunits of the V-ATPase complex (ATP6V1B1/*ATP6V1B1* and ATP6V1B2/*ATP6V1B2*), when compared to the untreated and the synchronized populations. On the other hand, lysosomal acid glycosylceramidase (GBA/*GBA)* and the Ras-related protein Rab-7a (RAB7A/*RAB7A*), which is involved in lysosome trafficking, revealed only very weak trends towards reduced levels in the mitotic cells as compared to the untreated and synchronized cells (Appendix A). On the contrary, we observed increased levels of mTOR and its phosphorylated form that may be indicative of mTORC1 or CDK1 activity and thereby suggest the downregulation of lysosomal biogenesis [4,8] (Appendix A). Indeed, the levels of the master regulator of lysosomal biogenesis, transcription factor EB (TFEB/*TFEB)*, were somewhat reduced in the released cells (Appendix A). It should, however, be noted that the level of phosphorylation of a classical mTORC1 substrate, ribosomal protein S6 kinase beta-1 (P70S6K/*RPS6KB1*), remained unchanged in all the samples (Appendix A).

To supplement the findings from the Western blot analyses, cathepsin B and D activities were measured in the different cell populations. While CTSD activity was similar in all the cell populations, CTSB activity was significantly increased in the released cell population (Appendix A). Since this result contradicted the lower levels of active forms of CTSB that was observed in the released cells, we compared the levels of cystatin-b (CSTB/*CSTB*), an endogenous cytosolic cysteine cathepsin inhibitor [52], in the synchronized and released cells (Appendix A). Multiple analyses revealed a strong trend for reduced CSTB levels in the released cells.

Taken together, these data indicate that the relative levels of several lysosomal proteins are reduced when cells enter mitosis.

### 3.3. Lysosomes Are Enlarged and Less Acidic during Mitosis

The observed downregulation of lysosomal proteins suggested that the volume of the lysosomal compartment was reduced during mitosis, possibly due to the reduced size of individual lysosomes or their decreased number. Comparing the numbers of lysosomes between the interphase and mitotic cells was not straightforward in 2D analysis due to the major difference in cross-sectional area of the cells. Instead, we examined the areas of individual lysosomes in the interphase and mitotic cells in asynchronous cell populations (Figure 2b). The average area of lysosomes in the mitotic cells was significantly larger than that of interphase cells (Figure 2c, Appendix A). Prompted by the increased lysosome size and above-described downregulation of active CTSB and the V-ATPase V1 subunit B that was involved in the luminal acidification in the mitotic cells, we next examined luminal pH of individual lysosomes in cells that were loaded with dextran coupled with pH-sensitive fluorescein (FITC) and pH-insensitive tetramethylrhodamine (TMR). The average intensity ratios of the two fluorophores in individual vesicles was significantly elevated during mitosis, indicating alkalization of lysosomes (Figure 2d). Concanamycin A, a V-ATPase inhibitor, was used as a positive control for lysosomal alkalization.

Taken together, these data indicate that the lysosomal area and pH are increased in mitotic cells.

### 3.4. Lipid Composition of Purified Lysosomes Are Different from Whole Cell

Next, we investigated whether and how the composition of lipids constituting the lysosomal membranes alters during mitosis. For this purpose, we combined the above-described protocol for preparing the mitotic cell population (Figure 1a) and a previously described protocol for lysosome purification that was based on magnetic capture of iron dextran (FeDEX)-loaded lysosomes [39]. This procedure yielded organelle fractions that were highly enriched for lysosomal proteins, including LAMP1, LAMP2, V-ATPase V0 subunit D1 (ATP6V0D1/*ATP6V0D1*), and CTSB, while depleted for proteins that were characteristic for other cellular compartments, including early endosomes (early endosome antigen 1 (EEA1/*EEA1*)), endoplasmic reticulum (protein disulfide-isomerase (PDI1/*P4HB*)), Golgi (Golgin-97/*GOLGA1*), mitochondria (Mitochondrial import receptor subunit TOM20 homolog (TOM20/*TOMM20*), and voltage-dependent anion-selective channel protein 1 (VDAC/*VDAC1*)) and chromatin (Histone H3 (Histone H3.1/*H3C1*) (Figure 3a).

We then analyzed the obtained lysosomal and whole cell fractions by quantitative MS-based shotgun lipidomics to quantitatively profile the lipids in these samples. Based on the measured mass to charge ratios (*m/z*) of the intact lipid ions and fragments that were generated from them, we identified an average of 356.2 (±11.4) lipid species in the lysosomal fractions and 355.2 (±5.7) in the whole cell fractions (Figure 3b). The arrest and release of cells had essentially no influence on the identified lipid species, as we identified nearly identical sets of lipid species in the three whole cell fractions and in the three lysosomal fractions, respectively (Figure 3c). On the contrary, we identified around 20 lipid species that were exclusively in the lysosomal fractions. Approximately 40% of these lipid species belonged to the lysosome-specific BMP/phosphatidylglycerol (PG) class. The identified lipid species belonged to 28 lipid classes in five lipid categories, fatty acyls (FA), glycerolipids (GL), glycerophospholipids (GPL), sphingolipids (SL), and sterol lipids (St) (Figure 3d). From here on, we treat the acyl-acyl GPLs (diacylGPL), alkyl/alkenyl-acyl GPLs (etherGPL), and mono-acyl/alkyl/alkenyl GPLs (lysoGPL) as separate categories.

Next, we quantified the identified lipid species using the determined ion intensities of the identified lipids and a series of internal lipid standards (Appendix A). The lysosomal fractions that were prepared from cells that were subjected to different treatments contained essentially the same total molar quantities of identified lipids (Figure 3e). Based on these total molar quantities, we calculated the yield of purified lysosomes to 1.33% (±0.31) of lipids that were present in the starting materials (Figure 3f).

We then calculated the molar percentages (mol%, molar quantities normalized to the total molar quantities of all the identified lipids) for all the identified lipid species to illuminate how the lipid compositions of purified lysosomes differed from those of the whole cells (Appendix A). In accordance with the observed depletion of mitochondrial proteins (Figure 3a), the mitochondrion-specific cardiolipin (CL), which accounted for approximately 3–4 mol% of lipids that were identified in the whole cell fractions, was nearly undetectable in the lysosomal fractions (Figure 3g). On the contrary, the purification enriched the lipid class BMP/PG from approximately 0.156 (±0.040) mol% in the whole cell fractions to 2.27 (±0.74) mol% in the lysosomal fractions (Figure 3h, Appendix A). Shotgun lipidomics cannot discriminate between the lysosomal BMP and the isomeric and mainly mitochondrial PG. Thus, we report them combined as BMP/PG). However, the BMP class is dominated by polyunsaturated species unlike the PG class, which is primarily composed of monounsaturated species [23]. Accordingly, the purification of lysosomes enriched mainly the polyunsaturated BMP/PG species but not monounsaturated species (Figure 3i). By employing the determined molar quantities of BMP/PG in the lysosomal and whole cell fractions as quantitative measures of lysosomes along the path of purification, we estimated that the purification recovered approximately 20% of the lysosomes that were present in the starting materials (Figure 3j).

Besides the above-described depletion of CL and the enrichment of BMP/PG, the prepared lysosomal fractions revealed distinctively different lipid profiles than the whole cell fractions. Linear modelling found 52.0% (±3.76%) of the identified lipid species at statistically significantly different levels in the lysosomal fractions compared to their respective whole cell fractions (Figure 3k). These belonged to all the seven lipid categories that were monitored in the analysis (Figure 3l). Plotting the fold changes (mol% in lysosomal fraction: mol% in the whole cell fraction) that were estimated for these species showed a clear trend of species of same lipid classes having similar values (Figure 3m). In accordance with this, the mol% values of the lipid classes in the lysosomal fractions greatly differed from those in the whole cell fractions (Figure 3m). The lysosomal fractions had reduced levels of diacylGPL classes except for the lysosome-specific BMP/PG and phosphatidylserine (PS), which is known for its enrichment at the plasma membrane (Figure 3m, heatmap). In contrast, all lysoGPL classes and the majority of sphingolipid (SL) classes were enriched in the lysosomal fractions. Besides the levels of lipid classes, the lysosomal fractions differed from the whole cell fractions in the quantitative profiles of the species constituting lipid classes (Figure 3m, plots to the right). We visualized such alterations by calculating the fold changes of the average numbers of carbon atoms and double bonds in species constituting individual lipid classes (numbers in the associated acyl, alkyl, alkenyl groups, and long-chain base). An overview of such fold changes that were calculated for all the monitored lipid classes illuminated the prominent reductions in the average numbers of acyl double bonds in diacylGPL classes. Taken together, these data indicate that the lysosome purifications yielded pure lysosome fractions and that the lipid composition of the lysosome fractions differ from that of the whole cell fractions.

### 3.5. The Lysosomal Lipid Profile Changes during the Cell Cycle

We next investigated whether and how the lipid profiles of lysosomes that were purified from the released cells differed from those from the control cells. The influence of the cell cycle arrest and release on the lipid profiles of the cells and lysosomes was evident in a correlation analysis. It showed that quantitative lipid profiles were significantly less similar between the samples that were derived from differently treated cells than between the biological replicas (Figure 4a, Appendix A). We applied linear modelling to identify the lipid species that were present at different levels in the samples that were derived from the released cells compared to corresponding controls (Figure 4b). The analysis found 132 and 109 altered species in the comparisons of lysosomal fractions of the released cells against those of the untreated and synchronized cells, respectively, and similarly 109 and 72 in comparisons of the whole cell fractions. The 55 lipid species found in both comparisons of lysosomal fractions primarily belonged to phosphatidylcholine (PC) and BMP/PG classes (Figure 4c,d). The 44 lipid species that were likewise identified in both comparisons of the whole cell fractions revealed, however, a distinct distribution pattern among the lipid classes (Figure 4c,d). This suggests that the alterations the lysosomal lipidome undergoes during mitosis is not simply a reflection of alterations that are occurring in the whole cell lipidome. Indeed, plotting the fold changes, estimated with linear modelling for lipid species, revealed distinctly altered patterns for the lysosomal fractions compared to the whole cell fractions (Figure 4e,f and Appendix A).

We noticed multiple lipid classes that were consistently up- or downregulated when comparing the lysosomal fractions from released populations with those from either untreated or synchronized populations (Figure 4e). These included the significantly down-regulated diacylGPL class PC and up-regulated SL classes SM and hexosylceramide (HexCer), all supported by multiple constituting species being regulated in the same direction (Figure 4e). Additionally, several lysoGPL classes including ether-linked lysophosphatidylcholine (LPC O-), lysophosphatidylserine (LPS), lysophosphatidylglycerol (LPG), and lysophosphatidylethanolamine (LPE) were increased in the lysosomes from the released cells compared to those from the untreated or synchronized cells (Figure 4e). Even though several classes were altered in the lysosomes from the released cells compared to those from the synchronized cells, the mol% of the entire lysoGPL category in lysosomes from the released and synchronized cells made up the same amount, which were in both cases significantly increased compared to the lysosomes from the untreated cells (Figure 4g, Appendix A). All the prepared lysosomal fractions were reduced for diacylGPLs in comparison to the whole cell fractions, in which diacylGPLs together accounted for >50 mol% of the identified lipids (Figure 4g). However, whereas diacylGPLs still accounted for 38.3 (±1.0) mol% and 34.1 (±3.0) mol% of lysosomal fractions of the untreated and synchronized cells, respectively, this value dramatically decreased to 28.3 (±1.9) mol% in those of the released cells (Figure 4g, Appendix A). This decrease owed largely to reduction in the level of the dominant PC class (Figure 4h). The average number of acyl double bonds in species of PC class likewise was decreased in the lysosomal fractions of the released cells (Figure 4i), whereas the same trend still was visible when including species of all the diacylGPL classes (except for BMP/PG with very high numbers of acyl double bonds) (Figure 4j). The observed decrease in the level of diacylGPL category in the lysosomal fractions of the released cells was accompanied by an increase in the levels of the SL category (Figure 4g, Appendix A) and of two major SL classes HexCer and SM (Figure 4k). We found none of the above-described lysosomal alterations in the whole cell fractions (Figure 4g–i, Appendix A), suggesting that they are caused by alteration in the lysosomal lipid metabolism or lipid translocations between the lysosomes and other cellular compartments.

The upregulation of SM was of particular interest, as inhibition of ASM and the resultant accumulation of SM in the lysosomes was previously demonstrated to be crucial for the CAD-induced lysosomal leakage [22]. To assert that the upregulation in lysosomal SM demonstrated here was not an artifact that was introduced by the applied technique of lysosome purification, we repeated the entire analysis by using another approach of lysosome purification employing an antibody against LAMP1 upstream of magnetic column chromatography. With this alternative approach, we recapitulated the same trends as those that were described above, even though the data were statistically weaker due to lower yields causing reduced data quality (Figure 4l, Appendix A). Importantly, the new approach, however, fully reproduced the above-described increase in the level of SM class in the lysosomal fraction of the released cells.

As the majority of our findings points towards reduced lysosomal membrane stability in mitotic cells, we next sought to test this hypothesis. To do so, asynchronous U2OS cells were treated with acridine orange, a metachromatic organic compound and a weak base with cationic properties. Acridine orange exhibits red fluorescence (561 nm laser) when accumulated in acidic compartments such as lysosomes, and green fluorescence (488 nm laser) when in a lower concentration outside the lysosome. Acridine orange is thought to induce peroxidative damage to lysosomal membranes upon exposure to blue light [53,54]. Whereas the interphase cells were able to withstand three minutes of exposure to blue light (488 nm) ultimately causing the characteristic “lysosomal explosion” phenotype without major morphological changes, mitotic cells displayed membrane blebs that were indicative of cell death more or less instantly after turning on the blue light (Figure 5). This finding demonstrates that mitotic cells are hypersensitive to lysosomal peroxidative damage supporting the hypothesis that mitotic lysosomes are more fragile than those of interphase cells.

## 4. Discussion

Lysosomal membrane integrity has been regarded to be essential for cell survival due to the degradative potential of the luminal hydrolases if released to the cytosol. Recently, several non-lethal roles for lysosomal leakage have surfaced (for more detail, see [24]) including mitotic lysosomal leakage, which occurs in close proximity to mitotic chromatin, where it assists the cell in completing accurate chromosome segregation [25]. Currently, very limited knowledge of lysosomal roles and changes during mitosis exists. Prompted by the involvement of lysosomes in mitosis, we wanted to investigate if changes in lysosomal composition occurs and if they could provide an explanation for the reduced lysosomal membrane integrity in mitosis. Here, we show, by using a carefully executed synchronization protocol yielding high amounts of mitotic U2OS cells, mitotic lysosomes greatly differ from those of interphase cells. By comparing mitotic lysosomes to those of interphase, we reveal that mitotic lysosomes are bigger, less acidic, and have significantly less lysosome-specific proteins and an altered lipid composition.

We demonstrate that mitotic cells have significantly reduced lysosomal proteins LAMP1 and CTSB compared to interphase (untreated and synchronized). To our understanding, this is the first time that mature CTSB has been investigated in mitotic cells. The decrease in mature CTSB could be explained by the increase in lysosomal pH that was observed in mitotic cells, since the acidic lumen is necessary for cathepsin maturation [55]. To our knowledge, there is only a single study that specifically investigates LAMP1 expression in mitosis. In this study, Yin and colleagues demonstrate that LAMP1 levels increase throughout the cell cycle from the G1 to the G2 phase due to lysosome biogenesis [56]. This discrepancy between our findings could be due to differences in the cell lines, quantification technique, or the selection of the loading control. Protein comparisons between the interphase and mitotic cells are complicated as the cells double the levels of the majority of their proteins and organelles during the S phase [57], whereas in mitosis, the transcriptional [29] and translational [30] activities are decreased. Furthermore, cells undergo extreme morphological changes during the cell cycle due to major cytoskeletal rearrangements that involve the commonly utilized loading controls tubulin and actin. It has been shown that tubulin levels increase 1.6−1.8-fold from G1 to mitosis, while the ratio between β-tubulin and the total protein ratio remains stable between the interphase and mitosis [58]. Here, we selected core histones (histone H3 or histone H2B) as the main loading control (Figure 2a, Appendix A), because of their total doubling during S phase of the cell cycle [59,60] ensures that the lysosomal protein levels are not exaggerated upon normalization. The loading controls should, however, be further investigated to find the most appropriate ones for protein comparisons between the cell cycle phases. Additionally, we report a significant increase in the mTOR protein level in mitotic cells compared to the untreated and synchronized cells, but mTORC1 activity that was measured by the phosphorylation of mTOR and its substrate P70S6K remain unchanged (Figure 2a, Appendix A). Others have demonstrated that mTORC1 activity that was analyzed by the phosphorylation of T389 of P70S6K, is downregulated during mitosis [8,34]. Additionally, Odle and colleagues show that due to CDK1-dependent phosphorylation of regulatory-associated protein of mTORC1 (RAPTOR/*RPTOR*) in mitosis, mTORC1 fails to localize to lysosomes [8], which is a necessary step in its activation [7]. As CDK1 is shown to regulate the mTORC1 pathway [8], we cannot rule out that the observed changes in mTOR and its substrates are caused by the CDK1 inhibitor that was used for the synchronization.

Contrary to the downregulation of mature CTSB protein levels, cathepsin B activity was increased in mitotic cells. This contradiction could be explained by the standard pH (pH = 6) that was used in the assay, which abolishes the effects of altered pH that were observed in living cells. Additionally, due to the lower levels of CSTB in the mitotic cells, cathepsin B activity was less inhibited in these samples when the cells are lysed and analyzed.

We demonstrate several interesting lipid changes occurring in mitotic lysosomes compared to untreated and synchronized ones, including significantly reduced diacylGPL class PC, and increased SL classes SM and HexCer, and trends of reduced lysoGPL classes LPC O-, LPS, LPG, and LPE. The purified lysosomes were enriched in BMP despite the generally low diacylGPL contents, in accordance with the specific localization of BMP at the intraluminal vesicles [17]. Additionally, the purified lysosomes containing high levels of SLs, cholesterol, and PS that are known to be rich at the plasma membrane likely reflected the heavy lipid trafficking from the plasma membrane to the lysosomes via endocytosis [61]. To our knowledge, this is the first time that lysosomal lipid changes during mitosis have been investigated. On a whole cell level, mitotic lipid changes have been implicated previously both on a single lipid basis by knockdown studies [26,27] and global lipidome changes between the interphase and mitosis [27]. In line with our results (Figure 4f, Appendix A), Artilla-Gokcumen et al. found that several ceramide species are increased in mitosis, but also that phosphatidic acid is increased 40-fold, whereas we see a decrease in phosphatidic acid species [27]. There are several differences between these studies including different cell lines, different synchronization reagents, and difference in comparisons (Atilla-Gokcumen and coworkers compared HeLa S-phase cells with cytokinetic cells), all of which could explain this discrepancy. Our lipidomics data suggest that lysosomes increase their SM contents during mitosis (Figure 4e). The elevation in SM and reduction in PC being solely observed in the purified lysosomes and not in the whole cell lysates, suggests that our observation reflects a local regulation in lysosomes. SM is mainly localized at the outer leaflet of the plasma membrane, and once endocytosed it ends up on the outer leaflet of the intraluminal vesicles facing the lumen of lysosomes [61]. We cannot, however, exclude that enhanced endocytosis during mitosis contributes to the altered lipid profile of lysosomes. The reduction of ASM levels in mitotic cells compared to the interphase cells (untreated cells (*p* = 0.0374) and synchronized cells (*p* = 0.1120), Figure 2a, Appendix A) was supported by the increase in SM. The reduction in ASM protein in the released cells was likely not a simple reflection of the overall lysosomal lipid metabolism shutting down, as the level of lysosomal acid glycosylceramidase (GBA/*GBA)* was not reduced in the released cells (Appendix A). The reduced ASM level and alkalized luminal pH that was found here thus may slow down SM degradation during mitosis. As lysosomal membranes contain high amounts of vitamin E, an antioxidant that protects the membrane lipids from oxidation, it would also be interesting to investigate whether a decrease in lysosomal vitamin E levels is associated with lysosomal membrane instability [62,63]. It should be noted however, that treatment with α-tocopherol or the α-tocopherol analogue trolox does not prevent mitotic lysosomal membrane permeabilization [25].

Taken together, we report here several novel findings of lysosomal changes in mitosis including the downregulation of proteins LAMP1, CTSB, and ASM, along with an increase in lysosomal area, and an increase in lipids SM and lysoGPLs. Interestingly, reduction of LAMPs [16], accumulation of SM [22] and lysoGPLs [23,64], and increase in lysosomal area [65,66] all promote lysosomal membrane instability.

This study further implicates lysosomal involvement in mitosis and provides a possible explanation to how selective controlled non-lethal lysosomal leakage occurs in mitosis to assist chromosome segregation.

## Figures and Tables

**Figure 1 cells-11-00875-f001:**
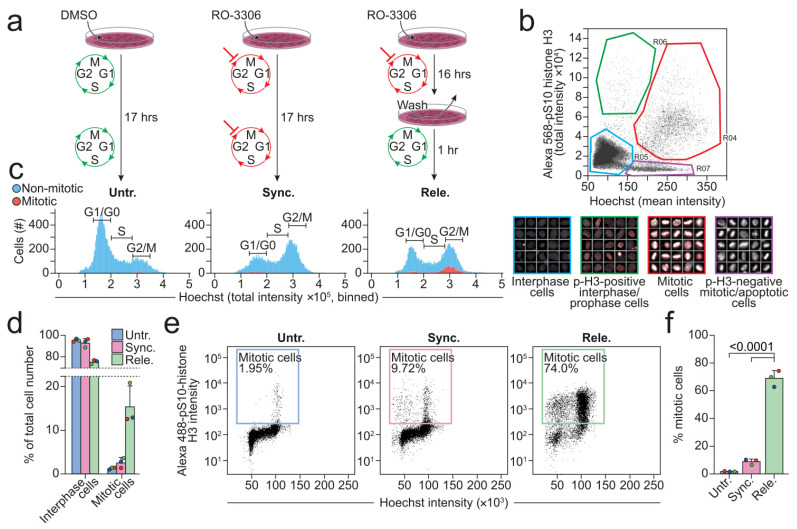
Procedure to enrich for mitotic cells. (**a**) An overview of the procedure that was used to prepare the untreated (vehicle), synchronized, and released U2OS cells. (**b**) Dot plot presenting the image-based measurement of the mean Hoechst and total p-H3 intensities of individual cells. The cells were categorized into four populations based on the color-coded gates in the dot plots: Interphase cells (blue), p-H3-positive interphase/prophase cells (green), mitotic cells (red), and p-H3-negative mitotic/apoptotic cells (purple). Randomly chosen cells from each gate are displayed beneath the dot plot in boxes that are framed by the same color as their respective gate. (**c**) A representative, image-based analysis of the cell cycle profiles, showing the numbers of mitotic (red) and non-mitotic (blue) cells having the values of total Hoechst intensity indicated on the x-axis. (**d**) Percentages of cells within interphase (blue) and mitotic (red) gate defined in (**b**). (**e**) Intensities of Hoechst and p-H3 in each cell was measured with flow cytometry. The gates display the mitotic cells as well as the percentage of cells within the gate. (**f**) The quantification of percentages of cells in mitosis as defined in (**e**). Circles with different colors (**d**,**f**) indicate the numbers that were obtained in each biological replicate. All the experiments were performed three times (n = 3). Statistical analyses were performed using unpaired *t*-tests. The *p*-values are stated in the respective figures. Abbreviation: p-H3, phospho-S10-histone H3.

**Figure 2 cells-11-00875-f002:**
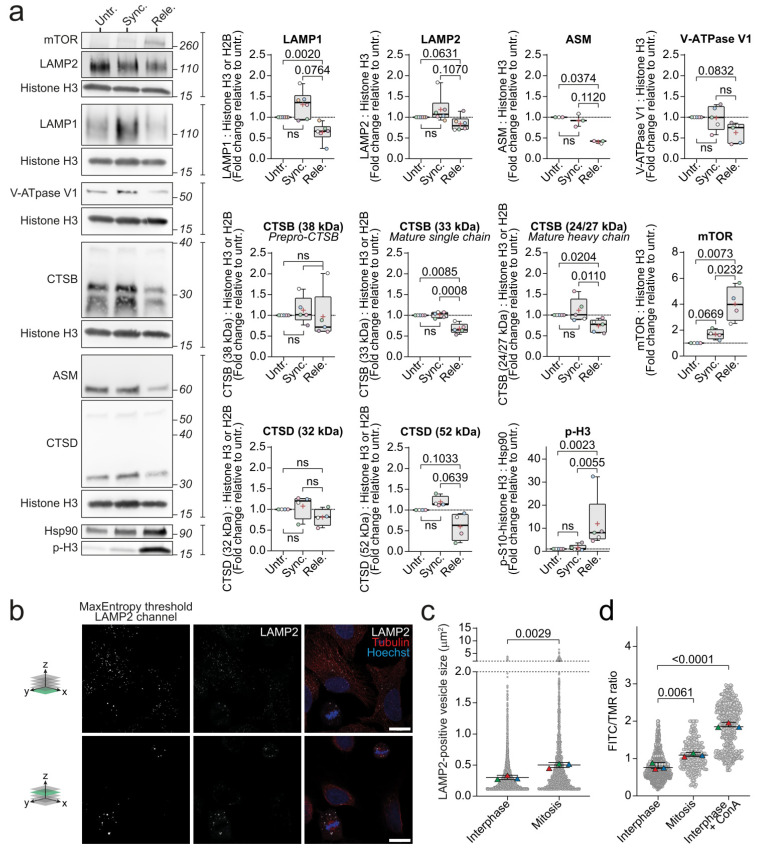
Mitotic cells decrease levels of lysosomal proteins, increase lysosome size, and alkalize lysosomal pH. (**a**) Representative immunoblotting of mitotic marker (p-H3) and lysosomal proteins in untreated, synchronized, and released U2OS cells and their respective quantifications of band intensities that were normalized to the respective loading controls, shown as fold change vs untreated. (**b**) Representative immunofluorescent images in two separate z-planes (illustrated by green z-plane on 3D axis to the left of images) of LAMP2 (middle and right row), α-tubulin (right row), and Hoechst (right row) in U2OS cells, and the binary image of the LAMP2 channel after employment of the automated max entropy threshold (left row). The scale bar is set to 10 µm. (**c**) Automated analysis of the areas of LAMP2-positive vesicles in asynchronous U2OS cells, defined by the threshold that was used in (**b**) and the particle size cutoff-value of 0.1 µm^2^-infinity. (**d**) The measured FITC/TMR intensity ratios of vesicles in interphase and mitotic U2OS cells. Triangles in different colors (**c**,**d**) indicate the values that were determined for each biological replicate. All the experiments were performed at least three times (n ≥ 3). Statistical analyses were performed on the average values using unpaired *t*-test. *P*-values are stated in the respective figures. Abbreviations: ASM, Acid sphingomyelinase; ConA, Concanamycin A; CTSB, Cathepsin B; CTSD, Cathepsin D; FITC, Fluorescein isothiocyanate; LAMP1/2, Lysosomal-associated membrane protein 1/2; mTOR, mammalian target of rapamycin; p-H3, p-S10-histone H3; TMR, Tetramethylrhodamine; V-ATPase, vacuolar-type ATPase.

**Figure 3 cells-11-00875-f003:**
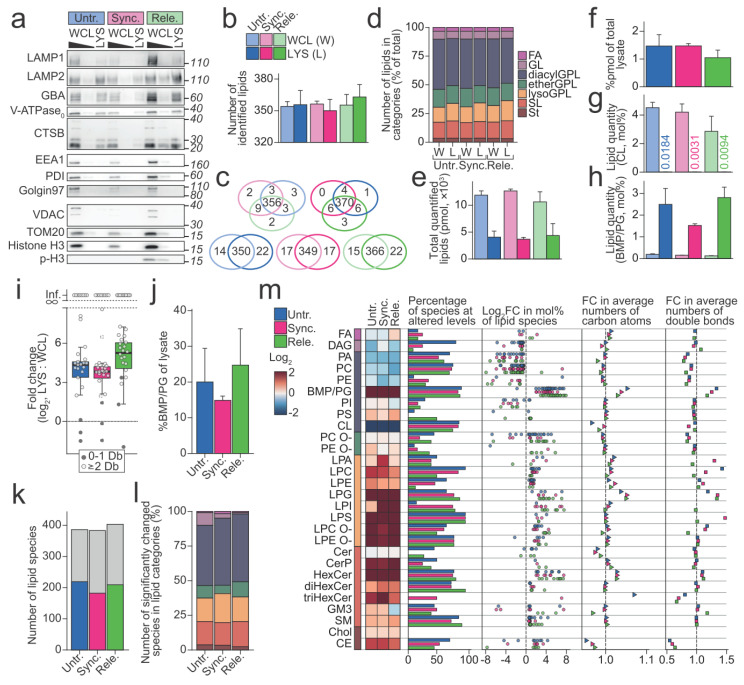
Lysosome purification and lysosomal lipid profiles. (**a**) Western blot of the whole cell lysates (WCLs) and lysosomal fractions (LYSs) from the untreated (Untr.), synchronized (Sync.), and released (Rele.) U2OS cells. LAMP1, LAMP2, GBA, V-ATPase V1 subunit (ATP6V1B1 and ATP6V1B2), and CTSB represent lysosomal proteins. EEA1 is a marker for early endosomes, PDI is a endoplasmic reticulum marker, and Golgin97 is a marker for Golgi apparatus. VDAC and TOM20 are markers for mitochondria and histone H3 and phosphorylated histone H3 (p-H3) represents the nuclear/chromatin fraction. The WCL samples are loaded in 5x and 1x, and the LYS samples are loaded in 1x based on the measured lipid contents. (**b**) The average number of lipid species in each sample type that were identified by quantitative mass spectrometry-based shotgun lipidomics. (**c**), Venn diagrams representing the number of lipid species that were identified in each sample type and shared among them. If a lipid species is identified in just a single replicate, it qualifies for being in the Venn diagram. The light colors represent the WCL samples, and the dark colors the LYS samples. Top: The untreated, synchronized, and released WCL samples (left) and the LYS samples (right). Bottom: WCL and LYS of untreated (left), synchronized (middle), and released cells (right). (**d**) The percentages of the identified lipid species belonging to the indicated lipid categories. (**e**) Total molar quantities of lipids in the analyzed aliquots of WCL and LYS. (**f**) Percentages of lipids in the total WCLs that were retained in the LYSs. (**g**) mol% of cardiolipin (CL) class in the WCLs and LYSs. (**h**) Mol% of bis(monoacyl)glycerophosphate (BMP)/phosphatidylglycerol (PG) in the WCLs and LYSs. (**i**) Fold change (log_2_) of BMP/PG species in LYSs relative to WCLs. Dots are colored according to number of acyl double bonds (Db). (**j**) The percentage of BMP/PG class in the total WCLs that were recovered in LYSs. (**k**) The number of identified lipid species (grey bars) and those that were found at statistically significantly different levels in LYSs in comparison to the respective WCLs (colored bars, *p*-value < 0.05) by linear modelling. (**l**), Distribution of the lipid species that were found at statistically significantly different levels in (**k**), in the indicated lipid categories. Color legend as in (**d**). (**m**) The lipid profiles of LYSs relative to the respective WCLs. From left to right: A heatmap of fold changes (log_2_) of the lipid classes; percentage of lipid species in each lipid class that was found at statistically significantly different levels by linear modelling; fold change (log_2_) estimated for the significantly changed lipid species; fold change of the average number of carbon atoms in each lipid class; fold change of average number of double bonds in each lipid class. All colors in the figure correspond to the legends in (**b**,**d**). All the experiments were performed three times (n = 3). *p*-values are stated in the respective figures. Abbreviations: BMP, bis(monoacyl)glycerophosphate; CTSB, Cathepsin B; diacylGPL, acyl-acyl glycerophospholipid; EEA1, Early endosome antigen 1; etherGPL, alkyl/alkenyl-acyl glycerophospholipid; FA, fatty acyl; GBA, β-glucocerebrosidase; GL, glycerolipid; LAMP1/2, Lysosomal-associated membrane protein 1/2; lysoGPL, mono-acyl/alkyl/alkenyl glycerophospholipid; PDI, Protein disulfide-isomerase; PG, phosphatidylglycerol; p-H3, p-S10-histone H3; SL, sphingolipid; St, sterol lipid; TOM20, Mitochondrial import receptor subunit TOM20 homolog; V-ATPase, vacuolar type ATPase; VDAC, voltage-dependent anion-selective channel protein 1. O- refers to ether-linked lipid classes.

**Figure 4 cells-11-00875-f004:**
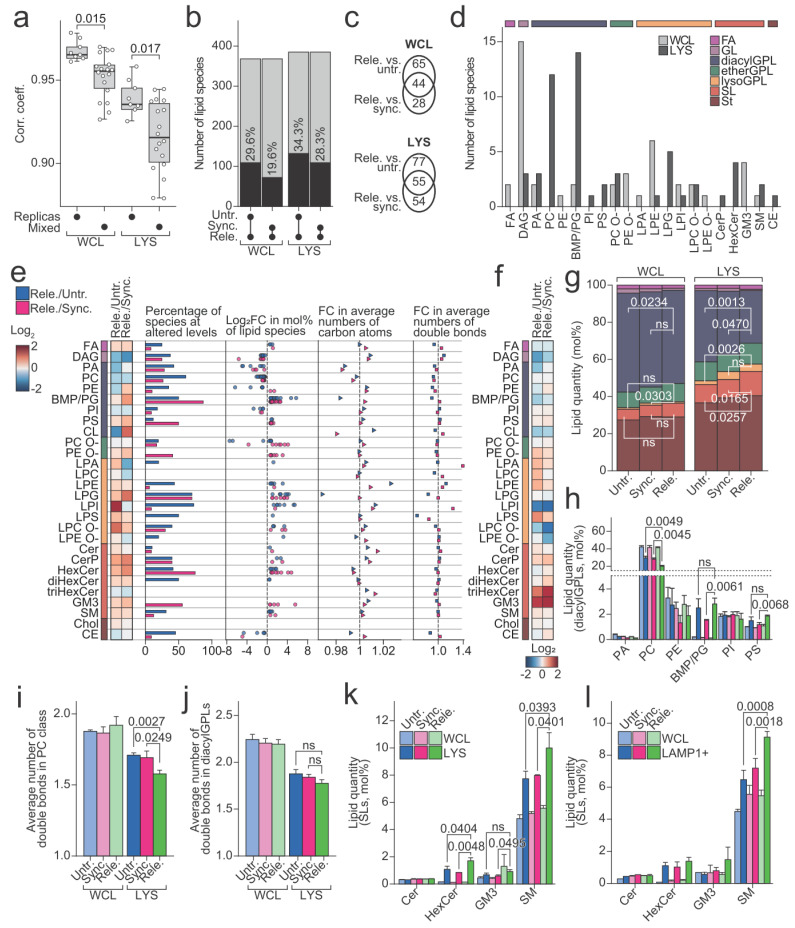
Unique lysosomal lipid profiles of mitotic cells. (**a**) Spearman correlation coefficients between the replicas or between the different sample types (mixed) that were calculated based on the acquired quantitative lipid profiles. (**b**) The number of identified lipid species (grey bars) and of those that were found at statistically significantly different levels in samples that were derived of released cells compared to the two control cells, by linear modelling (black bars, *p*-value < 0.05). (**c**) Venn diagrams representing the number of lipid species that were found at statistically significant different levels in the indicated comparisons, by linear modelling. (**d**) The number lipid species at statistically significant different levels in both comparisons of lysosomal fractions (LYSs) (dark grey) or of the whole cell lysates (WCLs) (light grey) belonging to the indicated lipid classes. (**e**) Lipid profiles of LYSs of the released cells relative to the controls. From left to right: A heatmap of the fold changes (log_2_) of lipid classes; percentage of lipid species in each lipid class that was found at statistically significant different levels by linear modelling; fold change (log_2_) that was estimated for the significantly changed lipid species; fold change of average number of carbon atoms in each lipid class; fold change of average number of double bonds in each lipid class. (**f**) Heatmap of the fold changes (log_2_) of lipid classes as in e, but for WCL. (**g**) The levels of lipid categories. Color legend as in e. (**h**) mol% of diacylGPL classes. (**i**) Average number of double bonds in PC class. (**j**) Average number of double bonds in diacylGPL category, excluding BMP/PG. (**k**) Levels of SL classes. (**l**) The levels of SL classes in WCL and purified LAMP1-positive compartment. All the experiments were performed three times (n = 3). *p*-values are stated in the respective figures. Abbreviations: Cer, ceramide; CL, cardiolipin; diacylGPL, acyl-acyl glycerophospholipid; etherGPL, alkyl/alkenyl-acylglycerophospholipid; FA, fatty acyl; GL, glycerolipid; GM3, ganglioside GM3; HexCer, hexosylceramide; lysoGPL, mono-acyl/alkyl/alkenyl glycerophospholipid; PA, phosphatidic acid; PC, phosphatidylcholine; PE, phosphatidylethanolamine; PI, phosphatidylinositol; PS, phosphatidylserine; SL, sphingolipid; SM, sphingomyelin; St, sterol lipid. O- refers to ether-linked lipid classes.

**Figure 5 cells-11-00875-f005:**
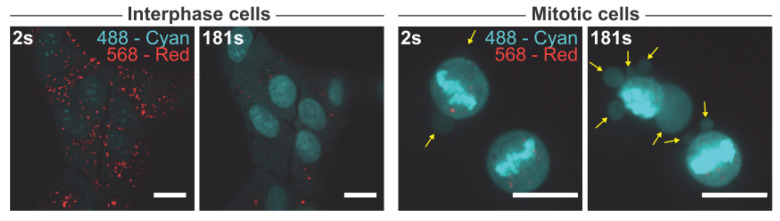
Mitotic cells are sensitive to lysosomal peroxidative damage. Representative images of acridine orange-treated interphase and mitotic cells. Cyan pseudo color displays the 488 nm channel and the 561 nm channel is shown as red. The scalebar is 20 µm. Cyan intensity is adjusted in the case of the mitotic cells to highlight the membrane blebs (yellow arrows) indicating the mitotic cells that are undergoing cell death. The experiment was performed multiple times using different concentrations of acridine orange and different laser settings, but at least two times with the settings used in the representative images. All the experiments had similar outcome.

## Data Availability

The data presented in this study are available from the corresponding author on request.

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
