# Peer review of "Lysosomal Changes in Mitosis"

_cells, 2022, doi:10.3390/cells11050875_

Round 1

Reviewer 1 Report

This is a nice and well-written manuscript, which indicates a landmark study of specific changes in lysosome in mitotic cells which may explain the reduced lysosome integrity observed during mitosis.  The authors showed significant differences in lysosome size, pH and composition between lysosomes isolated from human U2OS osteosarcoma cells either in mitosis or in interphase. Overall, the topic is timely and important and the approaches are logic to answer the questions.

However, I have some suggestions that could improve the impact of this manuscript:

Line 117: Isolation of mitotic cells was done through mitotic shake-off by knocking the dish into a hard surface. How do the authors discriminate mitotic cells from apoptotic cells?  

Figure 1C (right panel): Information is missing:  please explain these figures in the text and the figure legend.

Figure 2a: It is unclear how the specific loading control were chosen for protein quantification. 

As a complementary data, it would be nice to have the respective Red ponceau staining or each condition to visualize and quantify a total protein. This would confirm the differences in lysosomal proteome observed between mitotic and interphase cells.

H2B immuno-blot is missing, please add.

Why Hsp90 was used as a loading control for p-H3?

Figure 2C: please explain in the text why you used ConA?

Line 316: The authors stated that “Similar to LAMP1, ASM protein levels in released cells were significantly reduced as compared to untreated cells “but LAMP1 protein levels is quite similar in released cells  compared to  untreated cells  (p=0.076) (figure 2a). Please rectify.

Line 336: CTSB activity was significantly increased in the released cell population while reduced CSTB protein levels was observed in these cells. This seems to be contradictory. Could you explain?

Line 334: You found that downregulation of lysosomal proteins during mitosis is associated with an increase of lysosomal area. This seems to be contradictory. Could you explain?

Figure 5: the authors showed that mitotic cells are sensitive to lysosomal damage. Please evaluate lysosomal membrane damage (e.g, Galectin (8 or 3) staining, cathepsin release, etc ) and cell death  (e.g., PI staining, caspase activity, etc, ) in mitotic cells by  using complementary  methods. This would confirm the hypothesis that mitotic cells are more prone to lysosomal damage than interphase cells.

Author Response

Please find the author comments in the attached word document "Author response to reviewer 1".

Reviewer 2 Report

In this paper, the authors revealed that an increase in the pH of lysosomes during mitosis results in a decrease in both amount and activity of enzymes such as cathepsins, and also changes in the lipids that make up the lysosomal membrane. This study provides a new point of view and the method of cell enrichment is fine, and the biochemistry of lipid properties is well analyzed. Therefore, I have only minor comments. Specific comments are as follows.

  1. The protein levels of some enzymes are decreasing, what is the molecular mechanism for this? It is interesting to note that these changes occur even though the division phase is a very short period of time (about 1 h). It is not reasonable for these phenomena to be controlled at the transcriptional level. Why do you think these changes occur? And what are the biological implications? These points should be discussed as well.

  1. The lipid composition of lysosomal membranes during mitosis is altered (Figure 4), which may be responsible for the increased susceptibility to ROS (Figure 5). At a subcellular level, lysosomal membranes contain the highest levels of vitamin E (Rupar et al., 1992, doi: 10.1139/o92-075). The localization of vitamin E in the phospholipid bilayer of cell membranes is very important for the prevention of lipid oxidation. What about the possibility that the vitamin E concentration in the lysosomal membrane decreases during mitosis? I think it is worth discussing.

  1. The middle of Figure 1a is confusing; did you treat the cells with RO-3306 for 16 hours, remove it, and then add new RO-3306 again for 1 h (meaning totally treated it twice)?

Author Response

Point-by-point responses to reviewers

Please note that the indicated line numbers of cited sections refer to the revised version of the manuscript with the “track changes”.

Reviewer 2:

The protein levels of some enzymes are decreasing, what is the molecular mechanism for this?

We speculate in the discussion (line 596-600) that one molecular mechanism behind the decrease of lysosomal enzymes is the alkalization of the lysosomal pH: The decrease in mature CTSB could be explained by the increase in lysosomal pH observed in mitotic cells, since the acidic lumen is necessary for cathepsin maturation [55].

It is interesting to note that these changes occur even though the division phase is a very short period of time (about 1 h). It is not reasonable for these phenomena to be controlled at the transcriptional level. Why do you think these changes occur? And what are the biological implications? These points should be discussed as well.

We agree with the reviewer that the length of mitosis likely does not allow transcriptional regulation. Additionally, the observed increase in lysosomal pH suggests that mitotic lysosomes are metabolically inactive, in line with the inactivity of macroautophagy during mitosis (Odle et al. 2020). Regarding the biological implications, we hypothesize that the observed lysosomal changes allow a subset of lysosomes to undergo membrane permeabilization during prometaphase as described in discussion line 670-677: “Taken together, we here report several novel findings of lysosomal changes in mitosis including downregulation of proteins LAMP1, CTSB, and ASM, along with an increase in lysosomal area, and an increase in lipids SM and lysoGPLs. Interestingly, reduction of LAMPs [16], accumulation of SM [22] and lysoGPLs [23, 62], and increase in lysosomal area [63, 64] all promote lysosomal membrane instability. This study further implicates lysosomal involvement in mitosis and provides a possible explanation to how selective controlled non-lethal lysosomal leakage occurs in mitosis to assist chromosome segregation.”

The lipid composition of lysosomal membranes during mitosis is altered (Figure 4), which may be responsible for the increased susceptibility to ROS (Figure 5). At a subcellular level, lysosomal membranes contain the highest levels of vitamin E (Rupar et al., 1992, doi: 10.1139/o92-075). The localization of vitamin E in the phospholipid bilayer of cell membranes is very important for the prevention of lipid oxidation. What about the possibility that the vitamin E concentration in the lysosomal membrane decreases during mitosis? I think it is worth discussing.

Thank you for this interesting idea. We agree with the reviewer that this is indeed worth discussing and the following have been added to the discussion (line 664-669): “As lysosomal membranes contain high amounts of vitamin E, an antioxidant that protects membrane lipids from oxidation, it would also be interesting to investigate whether a decrease in lysosomal vitamin E levels is associated with  lysosomal membrane instability [62, 63]. It should be noted however, that treatment with α-tocopherol or the α-tocopherol analogue trolox does not prevent mitotic lysosomal membrane permeabilization [25].”

The middle of Figure 1a is confusing; did you treat the cells with RO-3306 for 16 hours, remove it, and then add new RO-3306 again for 1 h (meaning totally treated it twice)?

We agree that this figure was confusing. To ease the understanding we have changed the figure accordingly. The Untr. and Sync. populations were only treated once for 17hrs.